# Semantic description of equipment and its controls in building automation systems

Ganesh Ramanathan
*Siemens AG*
Zug, Switzerland
ganesh.ramanathan@siemens.com

Dr. Maria Husmann
*Siemens AG*
Zug, Switzerland
maria.husmann@siemens.com

## I. INTRODUCTION

Building automation (BA) systems orchestrate and monitor the functioning of a wide variety of utilities in a building so that living spaces are kept comfortable, safe, and secure. The complexity of such a system which involves multiple disciplines (heating, air-conditioning, lighting, fire safety, security etc.), coming from multiple vendors, is compounded by the fact that each building differs in the way the equipment operate and coordinate.

So far, efforts involving semantic modeling of BA systems have been focusing on the description of the control programs and the description of equipment, but have largely treated these two aspects separately. In fact, the *physical process* in which the equipment plays a role is missing in both of these aspects. As a result technical operators and service technicians had to understand the working of the system by piecing together information from different sources. Consider a room thermostat for example - just by knowing that it is wired to a temperature sensor and a radiator valve, it is not difficult to guess that the valve is controlled based on the room temperature. This intuition is built upon a number of facts in our mental knowledge base - like, that the valve on the radiator controls the amount of hot water flowing into it, and consequently, the amount of energy transferred by the radiator to air, whose temperature in turn is measured by the sensor. However, with large diversity of applications and size of deployments, such off-band interpretations become overwhelming. On the other hand, we realize that a semantic description which comprehensively describes the equipment, the process, and its control logic will enable us to create software agents for applications like fault detection and energy management. Though ontologies like Haystack [3], IFC [2], Brick [1], etc. attempt to label or classify equipment kinds, they do not approach the relation to control and process aspects. On the other hand, standards like ISO 16484 [5] assist in creating only comprehensive human-readable descriptions. From our experience at the Building Products division of Siemens AG we realize that the most detailed source of knowledge about the system originates from the engineers who design and commission the automation application in the building. In order to allow them to express their knowledge using supportive tools, we are creating a comprehensive BA ontology that captures the concepts of not only the automation aspect, but also the equipment and processes in our domain in an *integral* manner - i.e. we want this knowledge to naturally permeate at all levels of engineering workflow. In the following sections, we will describe the key use cases, the challenges in creating the ontologies, and the approach we adopted.

## II. USE CASES FOR A HOLISTIC SEMANTIC DESCRIPTION

### A. Fault detection and diagnostics

Fault detection and diagnostic (FDD) methods for building automation systems rely on rules which are based on the knowledge of the process, the control strategy, and the associated sensors and actuators. In the absence of a semantic description of such knowledge, we have to rely on proprietary conventions and off-band information to identify elements in the automation system which will provide inputs for processing the rules . Consider, for example, rules like the one provided by NIST [6] to recognize faults in air-handling units. They refer to process data using terms like "hot water temperature to the heating coil whose valve is regulated based on room temperature". To evaluate such rules, we wish to locate the data sources in our automation device without having to adopt proprietary methods like naming schema or syntactic tagging of the data points.

### B. Process-oriented knowledge

Technical operators are often interested in how the physical processes are organized and its automation works. For example, the operator might want to look at all oil-fired boilers in the building which have a pressurising pump and wants to know how its speed is regulated. The knowledge of the influence that the control has on the physical process therefore would not only detect anomalies, but also explain the state of the system. Taking the example of a heating system, in addition to the statement about a feedback control loop that takes a hot water temperature as input and drives an oil valve as output, we would like to describe that the consequence of this control loop is to maintain the supply temperature in the distribution header.

## C. Understanding process coordination

Apart from understanding the functioning of an equipment on its own, the coordination and dependencies at system level is also important. Consider, for example, the supply of hot water from boilers to different radiators across rooms. In order to optimize the response and efficiency of the system, demand, supply, and distribution need to be coordinated. Parts of this may even be dynamically reconfigured.

## III. CHALLENGES

In the description of the use cases above, we point out the need to describe the equipment, process, and controls in a comprehensive manner so that they weave seamlessly into our engineering process. When we started with the analysis to create such knowledge base in building automation domain, we encountered the following constraints:

- In the layered architecture of building automation systems the lower layers of field and automation devices are engineered based on detailed knowledge of the equipment and process they deal with. However, this knowledge is not made available as semantic data. Additionally, the tools and information modeling at each layer are diverse and manual reconstruction of the knowledge base as an option is economically unviable.
- Unlike the approaches where input and output elements in the control program are linked to sensor and actuators (like in [7]), we also need to link functions and parameters to both equipment components and process elements. Hence, the description of both the equipment and the process needs to be available when the control program is engineered.
- The BA industry lacks well-accepted ontologies to describe equipment, processes, and controls which can encourage its integration in engineering tools across manufacturers. However, we also do not aim for industry-wide monolithic ontologies which are centrally governed and hence will potentially pose a restriction on our applications that are evolving fast (like in smart energy).

## IV. APPROACH

We aimed for small iterations of experimentation with concepts and ideas so that we are able to demonstrate and validate it in real-life usage. We deconstructed the problem in to the following aspects:

- Avoid monolithic ontology by breaking down the knowledge bases to standardize abstract concepts and yet allow extensions in product- or discipline-specific concepts. Simultaneously, work on creating and supporting industry-wide standards to describe construction of equipment and processes.
- Create an ecosystem where engineering tools can access, use, and refer to the concepts as an integral part of the engineering workflow. This will enable bottom-up specification and extraction of knowledge where the control and construction aspects are inherently coupled.

- Build tools that enable authors of control applications to capture their knowledge using semantic data technologies.
- *Things* like sensors, actuators, and controllers need to be integrated as first-class citizens in the semantic description. In our experience, this integration is seamlessly achieved by using the Web of Things semantic *Thing Descriptions* [4].

## V. IMPLEMENTATION

Keeping the aspects previously highlighted as our guideline, we have developed some key ontologies and tools while validating them continuously in product usage and proof-of-concept implementations. We summarize the parts and the learning derived from them:

- As a pattern, we have adopted three layers of abstraction in the ontologies - beginning with *domain-level* concepts, which are then referred-to and specialized by *discipline-specific* (like HVAC, fire safety, lighting etc.) knowledge, and finally permit the flexibility of extensions in *product or system-specific* ontologies. Each of these layers is also vertically partitioned according to the concepts it deals with. For example, at the *domain-level* we talk about location, abstract functions and programs, assets, etc.
- We extended our primary engineering tool to generate semantic description of the building hierarchy and its contained automation systems. It is however still lacking the ability to describe the process.
- In order to link and hence provide semantic interoperability with other knowledge sources in the industry we designed the abstraction in manner that the terms can be bridged at the *domain-* and *discipline-specific* layers.
- We implemented a proof-of-concept demonstrator of semantic data-driven fault detection system which uses the semantically described engineering data to match them with fault detection rules (which are also described semantically) and allows the detection execution code to locate data points of interest without using any proprietary binding concepts.
- We are working on tools to extract knowledge from sources which are not explicitly machine readable - for example, process diagrams or text description of equipment functioning.

## VI. SUMMARY

We have argued that a bottom-up description in building automation should involve both the automation and the construction aspects in order to facilitate software agents like fault detection to reason about the functioning of the system. This requires a flexible and extensible knowledge base which is product-agnostic and yet open to linking against industry-wide ontologies. When such knowledge bases are made available to the engineering tools, it enables the domain expert to create a comprehensive description of the system.

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
