# OpenReview forum: "Semantic description of equipment and its controls in building automation systems"
_eswc-conferences.org/ESWC/2021/Conference/Industry_Track — Submitted to ESWC 2021 Industry_

### Official Review · ~Antoine_Zimmermann1 · 2021-04-16
**A relevant use of semantic technologies for a relevant problem, but a somewhat vague description of it**

**Rating:** 6
**Confidence:** 3

**Review:**

The paper argues that having an explicit, semantic description of a building infrastructure and associated resources (sensors, actuators) as well as their functionning (including prcesses and rules), greatly facilitates the development of building automation, especially in an interoperable way. The paper describes a few use cases, challenges, approach and implementation made at Siemens in relation to this problem.

Overall, the problem, the advocated approach solution, are relevant and meaningful for ESWC. However, I believe that much of the content is quite high level and generic and could have been replaced or rephrased by more specific and more precise description. As an example among many: "Build tools that enable authors of control applications to capture their knowledge using semantic data technologies." This is a very vague assertion, applicable to almost any problem. Another example "Avoid monolithic ontologies ..." -> arguing in favour of modularity, which is good, is one thing; managing to find the sweet spot between separation of concerns and cohesion is another. It is good to claim that best practices will be followed, but in such a submitted paper, we expect to see the means by which the practices are achieved.
Even in the "implementation" section, we are left with little vision of what works and what does not.

All in all, I would say that this paper can be accepted for presentation only if there is enough room for it.


Minor comments:
 - Section II-A.:
  * "for processing rules ." -> remove space before dot
  * ”hot water temperature to the heating coil whose valve is regulated
based on room temperature” -> the quotes at the beginning are closing quotes
 - Sec.II.B.
  * "pressurising" -> pressurizing*
  * "distribution header" -> distribution heater?
  * references: "George E Kelly" -> George E. Kelly and Steven T. Bushby

---

### Official Review · ~Anna_Himmelhuber1 · 2021-04-16
**Good outline on how to approach a holistic semantic description**

**Rating:** 7
**Confidence:** 3

**Review:**


The authors point out the highly relevant shortcomings of current semantic modeling of BA systems, which only either focus on the description of the control programs or the equipment, and completely missing the corresponding physical processes, which requires the user to fill this gap with mental models. Their approach is targeted at handeling this missing link by integrating knowledge about the system from the design engineers. The given use cases include fault detection and diagnosis as well as process-oriented knowledge.

The authors give a good summary of the challenges in creating a holistic semantic description and show ways how to handle them by abstracting them to a three layer process.  A sketch on how this is used on one of the use cases would have been a nice to have.

---

### Official Review · ~Konrad_Diwold1 · 2021-04-21
**Review "Semantic description of equipment and its controls in building automation systems"**

**Rating:** 3
**Confidence:** 5

**Review:**

In their article, the authors discuss how a holistic semantic description can be achieved in the context of building automation. Unfortunately, the submission is too long as it was handed in in IEEE format and not springer and would thus exceed the page limit and is very vague.

Most of the article (full first page) describes the initial situation of the semantic modeling in BA and the use-cases in mind.  What would be really interesting for the reader (Approach and Implementation) is presented really vaguely (imo too vaguely)

"IV Approach"
Here the requirements towards their approach are outlined rather than the approach itself. e.g., the authors write "Avoid monolithic ontology by breaking down the knowledge bases to standardize abstract concepts and yet allow extensions in product- or discipline-specific concepts. Simultaneously, work on creating and supporting industrywide standards to describe construction of equipment and processes" -

how do they want to achieve this / have they designed such a system? How does the design differ to similar existing approaches in this context e.g., Mayer, Simon, et al. "An open semantic framework for the industrial Internet of Things." IEEE Intelligent Systems 32.1 (2017): 96-101. ?

"V Implementation"
is also rather descriptive and vague - as a reader one asks: how was the PoC implemented?, what was the test setup?,  what are the learnings from an industry point of view?, is the system in use - planned to go in use?, are their more details on the key ontologies?

They for example write "We are working on tools to extract knowledge from sources which are not explicitly machine readable - for example, process diagrams or text description of equipment functioning." - this is extremely interesting, but would require more details to get merit out of it as a reader.